# YBX1 Regulates Satellite II RNA Loading into Small Extracellular Vesicles and Promotes the Senescent Phenotype

**DOI:** 10.3390/ijms242216399

**Published:** 2023-11-16

**Authors:** Masatomo Chiba, Kenichi Miyata, Hikaru Okawa, Yoko Tanaka, Koji Ueda, Hiroyuki Seimiya, Akiko Takahashi

**Affiliations:** 1Division of Cellular Senescence, The Cancer Institute, Japanese Foundation for Cancer Research, Tokyo 135-8550, Japan; masatomo.chiba@jfcr.or.jp (M.C.); kenichi.miyata@jfcr.or.jp (K.M.); hikaru.okawa@jfcr.or.jp (H.O.); y.tanaka@jfcr.or.jp (Y.T.); 2Department of Computational Biology and Medical Sciences, Graduate School of Frontier Sciences, The University of Tokyo, Tokyo 113-8654, Japan; hseimiya@jfcr.or.jp; 3Project for Cancer Epigenomics, The Cancer Institute, Japanese Foundation for Cancer Research, Tokyo 135-8550, Japan; 4Cancer Proteomics Group, Cancer Precision Medicine Center, Japanese Foundation for Cancer Research, Tokyo 135-8550, Japan; koji.ueda@jfcr.or.jp; 5Division of Molecular Biotherapy, Cancer Chemotherapy Center, Japanese Foundation for Cancer Rsearch, Tokyo 135-8550, Japan; 6Cancer Cell Communication Project, NEXT-Ganken Program, Japanese Foundation for Cancer Research, Tokyo 135-8550, Japan

**Keywords:** satellite II RNA, cellular senescence, EV, SASP, YBX1

## Abstract

Senescent cells secrete inflammatory proteins and small extracellular vesicles (sEVs), collectively termed senescence-associated secretory phenotype (SASP), and promote age-related diseases. Epigenetic alteration in senescent cells induces the expression of satellite II (SATII) RNA, non-coding RNA transcribed from pericentromeric repetitive sequences in the genome, leading to the expression of inflammatory SASP genes. SATII RNA is contained in sEVs and functions as an SASP factor in recipient cells. However, the molecular mechanism of SATII RNA loading into sEVs is unclear. In this study, we identified Y-box binding protein 1 (YBX1) as a carrier of SATII RNA via mass spectrometry analysis after RNA pull-down. sEVs containing SATII RNA induced cellular senescence and promoted the expression of inflammatory SASP genes in recipient cells. YBX1 knockdown significantly reduced SATII RNA levels in sEVs and inhibited the propagation of SASP in recipient cells. The analysis of the clinical dataset revealed that YBX1 expression is higher in cancer stroma than in normal stroma of breast and ovarian cancer tissues. Furthermore, high YBX1 expression was correlated with poor prognosis in breast and ovarian cancers. This study demonstrated that SATII RNA loading into sEVs is regulated via YBX1 and that YBX1 is a promising target in novel cancer therapy.

## 1. Introduction

Cellular senescence is a state in which the cell cycle is irreversibly arrested, preventing abnormal cell proliferation caused by the activation of oncogenes [1]. Therefore, cellular senescence inhibits cancer development. Conversely, senescent cells secrete senescence-associated secretory phenotype (SASP) factors, including inflammatory cytokines, chemokines, matrix-degrading enzymes, growth factors, and extracellular vesicles (EVs), which affect surrounding cells, leading to age-related diseases, including cancers [2,3,4,5,6,7]. Recent studies reported that cellular senescence is induced in normal cells, cancer-associated fibroblasts (CAFs), and a part of cancer cells in the cancer microenvironment [1,2,8,9,10,11,12].

The expression of inflammatory SASP genes is induced by the activation of some transcription factors such as nuclear factor-kappa B (NF-κB) and CCAAT/enhancer binding protein beta (C/EBPβ). Furthermore, epigenomic alteration is also implicated in the expression of SASP factors in senescent cells. We recently found that the DNA structure of the satellite II (SATII), repetitive sequences in the pericentromeric region of chromosomes, is changed during cellular senescence and non-coding RNA transcribed from the SATII region bound to CCCTC-binding factor (CTCF) [13,14]. Therefore, SATII RNA inhibits the function of CTCF and changes the chromatin interactions in some SASP gene loci, resulting in the expression of inflammatory SASP genes.

Mammalian cells secrete EVs of various sizes that act as intercellular communication tools [15,16]. EVs transmit proteins, lipids, nucleic acids, and metabolites to surrounding cells and induce phenotype changes such as cancer development [17,18,19].

We demonstrated that the secretion of small EVs (sEVs) is dramatically activated in senescent cells compared to normal cells [7,20,21,22]. Moreover, SATII RNA in sEVs from senescent cells enhances chromosomal instability and the expression of inflammatory genes in recipient cells [13,14]. Since SATII RNA expression is upregulated in most cancer and senescent stromal cells in the cancer microenvironment, SATII RNA in sEVs may contribute to the malignant phenotype of cancers.

A previous report demonstrated that major satellite RNA, the mouse orthologue of pericentromeric SATII RNA, binds to Y-box binding protein (YBX1) and inhibits YBX1-mediated DNA repair, thereby contributing to cancer progression [23]. YBX1 is a multifunctional protein that binds DNA, RNA, and proteins and is involved in various intracellular functions such as transcriptional regulation and cell proliferation [24]. It also transports various ncRNAs into sEVs [25,26,27]. However, it is not clear whether SATII RNA is transferred into sEVs via YBX1 in senescent cells.

In this study, we identified YBX1 as the agent of SATII RNA uptake into sEVs. YBX1 knockdown decreased SATII RNA levels in sEVs secreted from senescent cells. Furthermore, these sEVs had a lower ability to induce SASP in recipient cells. These results suggest that YBX1 contributes to cancer malignancy by regulating SATII RNA loading into sEVs in the cancer microenvironment.

## 2. Results

### 2.1. YBX1 Selectively Binds to SATII RNA

In previous our study, SATII RNA function that binds and inhibits CTCF was revealed using SVts-8 [13]. We performed a mass spectrometry analysis of binding proteins after the RNA pull-down of SATII RNA using lysates of SVts-8 cells to identify proteins involved in the translocation of SATII RNA to sEVs (Figure 1A). Gene ontology (GO) term analysis showed that RNA-binding proteins were significantly enriched (Figure 1B). Among the top 20 enriched RNA-binding proteins, YBX1, heterogeneous nuclear ribonucleoprotein A (HNRNPA), and heterogeneous nuclear ribonucleoprotein K (HNRNPK) were detected, which have been associated with sEVs (Figure 1C). YBX1 specifically binds to major satellite RNA, the mouse orthologue of pericentromeric non-coding RNA [23]. Based on this previous knowledge, we, therefore, further investigated YBX1. SATII RNA is transcribed from repetitive sequences located in the pericentromere region, while SATα RNA is transcribed from repetitive sequences located in the centromere of the genome. Therefore, we set SATα RNA as the control. Western blot analysis after RNA pull-down demonstrated that SATII RNA, but not SATα RNA, bound to YBX1 in both nuclear and cytoplasmic fractions of lysates from SVts-8 cells (Figure 1D). These results suggest that YBX1 binds selectively to SATII RNA.

### 2.2. YBX1 Incorporates SATII RNA into sEVs

To confirm whether YBX1 is involved in the translocation of SATII RNA into sEVs, we prepared two types of senescent cells, X-ray irradiation-induced senescence (IR) in retinal pigment epithelial-1 (RPE-1) cells and doxorubicin (DXR)-induced senescence in IMR-90 cells. Senescence induction was confirmed via senescence-associated β-galactosidase (SA-β-gal) staining. SA-β-gal positive cells significantly increased with IR or DXR treatment (Figure 2A and Appendix A). In addition, we observed a decrease in the levels of laminB1 gene (LMNB1) mRNA and an increase in the RNA levels of CDKN1A and SATII RNA levels in senescent cells, as previously reported (Figure 2B and Appendix A) [13]. Interestingly, YBX1 expression was upregulated in senescent cells (Figure 2B). We also detected the increased expression of SASP-related genes such as *IL-1A* (Interleukin-1 alpha), *IL-1B* (Interleukin-1 beta), *IL-6* (Interleukin-6), *IL-8* (Interleukin-8), and *INFB1* (Interferon-beta 1) (Figure 2B and Appendix A). The percentage of cells showing co-localization of the DNA damage markers γH2AX and pST/Q increased after induction of cellular senescence (Figure 2C and Appendix A).

We performed YBX1 knockdown on these senescent cells and confirmed that the expression levels of YBX1 mRNA and protein amount were significantly reduced (Figure 3A,B and Appendix A). Next, we verified the characteristic changes of sEVs via YBX1 knockdown. The number and particle size of sEVs derived from YBX1 knockdown or control cells were measured using a nanoparticle analyzer, and no significant differences were found (Figure 3C and Appendix A). Immunoelectron microscopy using an anti-CD63 antibody, one of the sEVs markers, revealed no differences in the particle shape of sEVs on YBX1 knockdown (Figure 3D). These data indicate that YBX1 knockdown had no effect on the number, size, or shape of the particles. Finally, to verify the selectivity of SATII RNA incorporation into sEVs by YBX1, the incorporation of SATII RNA or SATα RNA into sEVs was examined via real-time quantitative polymerase chain reaction (PCR). The amount of SATII RNA in sEVs decreased after the YBX1 knockdown, but the amount of SATα RNA in sEVs did not show any significant changes (Figure 3E and Appendix A). These results suggest that YBX1 regulates SATII RNA incorporation into sEVs regardless of the cell type or senescence induction pathway.

### 2.3. Small EVs Derived from Senescent Cells Promote Senescent Phenotypes in Normal Cells via SATII RNA Transferred by YBX1

To determine the functions of sEVs containing SATII RNA translocated via YBX1 on surrounding cells, we examined the effects of sEVs collected from YBX1 knockdown or control conditions (Figure 4A). First, we found that the number of SA-β-gal–positive cells increased in the cells treated with sEVs collected from X-ray-induced senescent RPE-1 cells compared with the cells treated with the phosphate-buffered solution (PBS) (Figure 4B). Moreover, sEVs collected from YBX1 knockdown cells significantly decreased the number of SA-β-gal-positive cells compared with sEVs collected from siCtrl cells (Figure 4B). Next, sEVs collected from X-ray-induced senescent RPE-1 cells were added to proliferating RPE-1 cells to assess the expression of SASP factors. We found that the expression of these inflammatory SASP factors (IL1A, IL1B, IL6, IL8, and INFB1) increased in the cells treated with sEVs collected from senescent cells but not in the cells treated with PBS (Figure 4C). However, inflammatory gene expression was significantly decreased in the cells treated with sEVs derived from YBX1 knockdown cells (Figure 4C). These results indicate that YBX1 regulates the promotion of senescent phenotypes in normal recipient cells via SATII RNA transportation into sEVs.

### 2.4. YBX1 Expression Correlates with Poor Cancer Prognosis

SATII RNA is involved in malignant transformation in cancer [13,14]. Therefore, we analyzed the expression of YBX1, which is responsible for the sEVs-mediated secretion of SATII RNA to the extracellular matrix, using datasets on breast and ovarian cancers. Interestingly, microarray data (GSE4823, GSE40595 [28,29]) showed that *YBX1* expression in breast and ovarian cancer stromal tissues was higher than that in normal stromal tissues (Figure 5A). Furthermore, clinical data from The Cancer Genome Atlas (TCGA) database showed that breast and ovarian cancer patients with high *YBX1* expression showed poor prognosis with shorter recurrence-free survival. These data strongly suggest the involvement of YBX1 in cancer pathogenesis (Figure 5B).

## 3. Discussion

In this study, we found that YBX1 selectively regulates SATII RNA incorporation into sEVs in senescent cells (Figure 6). We demonstrated that YBX1 selectively binds to SATII RNA but not to SATα RNA in a sequence-dependent manner [30]. However, SATII RNA does not have YBX1 binding sequences. Further investigation is warranted to understand the binding mechanism of YBX1 to SATII RNA.

Our data show that sEVs addition collected from YBX1 knockdown cells dampens the upregulation of SASP gene expression such as *IL-1α*, *IL-6*, and *IL-8*. IL-1α enhances cancer progression by upregulating the expression of the SASP factor [31,32]. IL-6 is an SASP factor that causes EMT-like gene expression and drug resistance in surrounding cells, leading to cancer progression [33,34,35,36]. IL-6 and IL-8 induce EMT and stem-like features in surrounding cells [37,38]. Recent studies are underway in investigating the suppression of age-related diseases by inhibiting the secretion of SASP factors (senomorphics) or inducing selective cell death of senescent cells (senolytics) [39,40,41,42]. In addition, sEVs secreted from senescent stromal and cancer cells contribute to cell-autonomous carcinogenesis and tumor promotion in the cancer microenvironment [6,43,44]. sEVs containing SATII RNA cause chromosomal instability and inflammatory gene expression in surrounding normal cells and affect scaffold-independent growth [11]. Therefore, the inhibition of SATII RNA contained in sEVs may suppress malignant cancer progression. SATII RNA in sEVs induces the senescence of surrounding normal and cancer cells and promotes the secretion of tumorigenic SASP proteins such as IL-6 [7,45]. The propagation mechanism of the senescent phenotype via sEVs observed in this study may upregulate the negative effects of SASP on cancer cells. Our data from the clinical data analysis demonstrate that *YBX1* expression is significantly higher in cancer stromal tissues (Figure 5A). Other investigations have shown that YBX1 is highly expressed in various cancer types, including non-small cell lung cancer and colorectal cancer [46,47,48]. Moreover, re-analysis of the database revealed that high *YBX1* expression is associated with poor prognosis in human breast and ovarian cancers (Figure 5B), suggesting that YBX1 promotes cancer progression via sEVs containing tumorigenic SATII RNA. Further analysis is needed to clarify the involvement of YBX1 in the pathogenesis of sEVs derived from senescent stromal cells. In summary, targeting YBX1 may inhibit SATII RNA expression in sEVs secreted from senescent stromal cells and provide a novel therapeutic strategy for cancer treatment.

## 4. Materials and Methods

### 4.1. Cell Culture

RPE-1 cells were obtained from clonetech (Tokyo, Japan) [13], and IMR-90 cells were obtained from ATCC (Manassas, VA, USA). RPE-1 and IMR-90 cells were cultured in Dulbecco’s Modified Eagle’s Medium (DMEM; Nacalai Tesque, Kyoto, Japan) supplemented with 10% fetal bovine serum (FBS) and 1% penicillin/streptomycin (Sigma-Aldrich, St. Louis, MO, USA) at 37 °C. SVts-8 cells were cultured in DMEM (Nacalai Tesque) supplemented with 10% FBS and 1% penicillin/streptomycin (Sigma-Aldrich) at 34 °C [49].

### 4.2. Cellular Senescence Induction

To induce cellular senescence via X-ray, RPE-1 cells were irradiated at 40 Gy using the CP-160 X-ray system (Gulmay, England, UK). After irradiation, RPE-1 cells were plated at a density of 4000 cells cm^2^ and were not passaged for 10 days after irradiation. IMR-90 cells were plated at a density of 4000 cells cm^2^ and treated with 200 ng/mL DXR to induce cellular senescence the next day. These cells were not passaged for 11 days after treatment.

### 4.3. RNA Interference

RNA interference was performed via transfection of siRNA oligos with Lipofectamine™ RNAiMAX transfection reagent (13778075; Thermo Fisher Scientific, Waltham, MA, USA) according to the manufacturer’s protocol. The sequences of siRNA oligos are as follows: siCtrl, GCGCUUGUAGGAUUCG; siYBX1 #1, GAGAGACUGGAGUUGA; siYBX1 #2, GCGGAGGCAGCAAAUGUUA.

### 4.4. Reverse Transcription Quantitative PCR

Total RNA was extracted from cultured cells using the mirVana miRNA Isolation Kit (AM1561; Thermo Fisher Scientific) and treated with TURBO DNase (AM2238; Thermo Fisher Scientific) to remove genome DNA. Extracted RNA was subjected to reverse transcription using the PrimeScript RT reagent kit (RR037A; TaKaRa Bio Inc., Tokyo, Japan). Reverse transcription quantitative PCR (RT- qPCR) was performed on the StepOnePlus PCR system (Applied Biosystems, Bedford, MA, USA) using SYBR Premix Ex Taq (RR820A, TaKaRa Bio Inc.). The PCR primer sequences used were as follows; human *actin β*, 5′-AGAGCTACGAGCTGCCTGAC-3′ (forward) and 5′-AGCACTGTGTTGGCGTACAG-3′ (reverse); human *LMNB1*, 5′-GGGAAGTTTATTCGCTTGAAGA-3′ (forward) and 5′-ATCTCCCAGCCTCCCATT-3′ (reverse); human *CDKN1A*, 5′-TCAGGGTCGAAAACGGCG-3′ (forward) and 5′-AAGATCAGCCGGCGTTTGGA-3′ (reverse); human SATII RNA, 5′-AATCATGGAATGGTCTCGAT-3′ (forward) and 5′-ATAATTCCATTCGATTCCA-3′ (reverse); human *IL-1α*, 5′-AACCAGTGCTGCTGAAGGA-3′ (forward) and 5′-TTCTTAGTGCCGTGAGTTTCC-3′ (reverse); human *IL-1β*, 5′-CTGTCCTGCGTGTTGAAAGA-3′ (forward) and 5′-TTGGGTAATTTTTGGGATCTACA-3′ (reverse); human *IL-6*, 5′-CCAGGAGCCCAGCTATGAAC-3′ (forward) and 5′-CCCAGGGAGAAGGCAACTG-3′ (reverse); human *IL-8*, 5′-AAGGAAAACTGGGTGCAGAG-3′ (forward) and 5′-ATTGCATCTGGCAACCCTAC-3′ (reverse); human *INFB1*, 5′-ACGCCGCATTGACCATCTAT-3′ (forward) and 5′-GTCTCATTCCAGCCAGTGCT-3′ (reverse); human *YBX1*, 5′-GGAGTTTGATGTTGTTGAAGGA-3′ (forward) and 5′-AACTGGAACACCACCAGGAC-3′ (reverse); human SATα RNA, 5′-AAGGTCAATGGCAGAAAAGAA-3′ (forward) and 5′-CAACGAAGGCCACAAGATGTC-3′ (reverse).

### 4.5. RNA Pull-Down Assay

RNA pull-down assays were performed using the RiboTrap Kit (#RN1011/RN1012; MBL, Tokyo, Japan) according to the manufacturer’s instructions. Briefly, 5-bromo-UTP was randomly incorporated into hSATα and hSATII RNA during transcription using a vector containing full-length RNA as the template. Next, anti-BrdU antibodies conjugated with Dynabeads Protein G (#10004D; Thermo Fisher Scientific) were bound to the RNA synthesized in vitro and incubated with SVts-8 cell lysates or RPE-1 cell lysates overnight at 4 °C. Finally, samples were washed, eluted, and subjected to Western blot analysis.

### 4.6. Mass Spectrometry

RNA pull-down samples were reduced via incubation with 1× Laemmli sample buffer containing 10 mM TCEP for 10 min at 100 °C. Alkylation with 50 mM iodoacetamide for 45 min at room temperature was followed by sodium dodecyl sulfate–polyacrylamide gel electrophoresis (SDS-PAGE). Electrophoresis was stopped at a distance of 2 mm from the top of the separation gel. The gel was stained with Coomassie Brilliant Blue, and the protein bands were cut out. The protein bands were then de-stained and cut prior to in-gel digestion in Trypsin/Lys-C Mix (Promega, Tokyo, Japan) for 12 h at 37 °C. Peptides were extracted from the gel fragments and analyzed with the Orbitrap Fusion Lumos Mass Spectrometer (Thermo Scientific) and UltiMate 3000 RSLC nano-flow HPLC (Thermo Scientific). Tandem mass spectrometry spectra were matched against the SwissProt *Homo sapiens* protein sequence database using Proteome Discoverer 2.2 (Thermo Scientific). Peptide identification filters were set to a false discovery rate < 1%. GO analysis was performed using David (https://david.ncifcrf.gov/home.jsp; accessed on 6 October 2023).

### 4.7. Western Blot Analysis

For Western blot, cells were incubated in lysis buffer (10 mM Tris-HCl [pH 7.5], 140 mM NaCl, 1 mM EDTA, 1% TritonX-100, 0.1% SDS, and 10 mM β-glycerophosphate) containing 1% protease inhibitor cocktail (25955-11; Nacalai Tesque). Protein concentrations were determined using the Pierce™ BCA Protein Assay Kit (#23225; Thermo Fisher Scientific), separated by SDS-PAGE, and transferred to polyvinylidene fluoride membranes (Merck Millipore, Burlington, MA, USA). After blocking with 5% milk (Megmilk Snow Brand Co., Ltd., Sapporo, Japan), membranes were probed with primary antibodies targeting YB1 (4202, 1:1000; Cell Signaling Technology, Danvers, MA, USA) and α-tubulin (T9026, 1:2000; Sigma-Aldrich). Membranes were incubated with mouse (NA931-1ML; GE Healthcare, Chicago, IL, USA) or rabbit secondary antibodies (NA934-1ML; GE Healthcare) and incubated with SuperSignal West Femto Maximum Sensitivity Substrate (34096; Thermo Fisher Scientific) and detected using FUSION SOLO S (Vilber Lourmat, Collegien, France).

### 4.8. Isolation of sEVs from Cells

For sEVs purification, FBS was ultracentrifuged at 100,000× *g* for 16 h to remove microvesicles. Conditioned medium (CM) was prepared by adding 5% FBS to DMEM. sEVs were obtained from the supernatant with slight modifications to the previously described procedure [45]. Briefly, after incubating cells in CM for 48 h, the supernatant was collected and centrifuged at 300× *g* for 5 min to remove cells, followed by centrifugation at 2000× *g* for 10 min to remove cell residue. The supernatant was further centrifuged at 7600 rpm using the Optima L-90K Ultracentrifuge (Beckman Coulter, Brea, CA, USA) for 30 min and filtered through a 0.22-μm pore filter (Sartorius Stedim Biotech, Göttingen, Germany) to remove contaminating apoptotic bodies, expelled vesicles, and cell residues. The resulting supernatant was further centrifuged at 26,500 rpm using the Optima L-90K Ultracentrifuge for 3 h and at 45,000 rpm twice using the Optima MAX-TL Ultracentrifuge (Beckman Coulter) for 70 min. The number and size of particles were determined via nanoparticle tracking analysis using the NanoSight LM10 system (Malvern Panalytical Ltd., Malvern, Germany) or ZetaView (Particle Metrix GmbH Inc., Bavaria, Germany).

### 4.9. Electron Microscopy

sEVs isolated from RPE-1 cells were absorbed to formvar carbon-coated nickel grids, followed by immune-labeled with an anti-CD63 antibody (556019, BD Biosciences, San Diego, NJ, USA) and incubated 5 nM of a gold-labeled secondary antibody (British BioCell International Ltd., Cardiff, UK). The samples were fixed with 2% glutaraldehyde in 0.1 M phosphate buffer and used 2% phosphotungstic acid solution (pH 7.0) for negative staining. The grids were incubated on 2% glutaraldehyde in 0.1 M phosphate buffer and dried. They were stained with 2% uranyl acetate for 15 min and lead stain solution (Sigma-Aldrich). The samples were observed using a transmission electron microscope (JEM-1400Plus, JEOL Ltd., Tokyo, Japan) at 80 kV. Digital images were obtained with a CCD camera (VELETA, Olympus Soft imaging solutions GmbH, Olympus, Tokyo, Japan) [21].

### 4.10. Application of Exosome-like EVs

For the addition of EVs to cells, the collected EVs were mixed with EV-depleted conditioned medium at a density of 2 × 10^9^ particles/mL, and the host cell medium was replaced with EV-containing medium daily for 1 week. RNA from EV-treated cells was collected as described.

### 4.11. SA-β-Gal Assay

Cells were fixed in fixation buffer (2% paraformaldehyde and 0.2% glutaraldehyde in PBS) and incubated in staining solution (5 mM potassium ferricyanide, 5 mM potassium ferrocyanide, 2 mM MgCl_2_, 150 mM NaCl, and 1 mg/mL X-Gal) in citrate/sodium phosphate buffer (pH 6) overnight at 37 °C. After staining, cells were washed twice with PBS, and the percentage of stained cells was determined.

### 4.12. Immunofluorescence Microscopy

Cells were fixed in 4% paraformaldehyde/PBS (163-20145; Fujifilm Wako Chemicals, Tokyo, Japan) and permeabilized with 0.2% Triton X-100/Tris-buffered saline for 5 min at room temperature. For blocking, cells were incubated with 1% bovine serum albumin (A3059; Sigma-Aldrich) and 10% goat serum (G9023; Sigma-Aldrich) in Tris-buffered saline at 4 °C for 1 h. Cells were then incubated with primary antibodies targeting γ-H2AX (05-636, 1:1000; Millipore, Darmstadt, Germany) and phosphor (Ser/Thr) ATM/ATR substrate (2851, 1:500; Cell Signaling Technology), followed by incubation with secondary antibody coupled to Alexa Fluor 488 or Alexa Fluor 594 (Thermo Fisher Scientific) and 4′,6-diamidino-2-phenylindole (342-07431; Dojindo, Tokyo, Japan) to stain nuclei. DNA damage-positive cells were quantified using a fluorescence microscope (Carl Zeiss, Oberkochen, Germany).

### 4.13. Data Acquisition

Breast and ovarian cancer cohorts, including GSE4823 and GSE40595, were downloaded from the Gene Expression Omnibus (GEO) databases [28,29]. Correlation of recurrence-free survival prognosis and YBX1 expression level in patients with breast cancer (GSE25066) or ovarian cancer (GSE30161) was analyzed using the Kaplan–Meier plotter (https://kmplot.com; Accessed on 5 September 2023). High- and low-expression patient groups were stratified according to optimal gene expression cutoff values.

### 4.14. Statistical Analysis and Reproducibility

Statistical analysis was performed using one-way analysis of variance coupled with an unresponsive two-tailed Student’s *t*-test or Dunnett’s multiple comparison test. Statistical analyses were performed using PRISM software version 7.04 (MDF Co., Ltd., Tokyo, Japan). *p*-Values < 0.05 were considered statistically significant. Error bars indicate mean ± standard deviation. Results were obtained at least three times unless otherwise indicated.

## 5. Conclusions

This study reveals that YBX1 regulates the selective incorporation of SATII RNA into sEVs in senescent cells. These sEVs promote the senescent phenotype in surrounding recipient cells in the cancer microenvironment. Thus, YBX1 may be a useful therapeutic target in cancer therapy to regulate tumorigenic SASP factors.

## Figures and Tables

**Figure 1 ijms-24-16399-f001:**
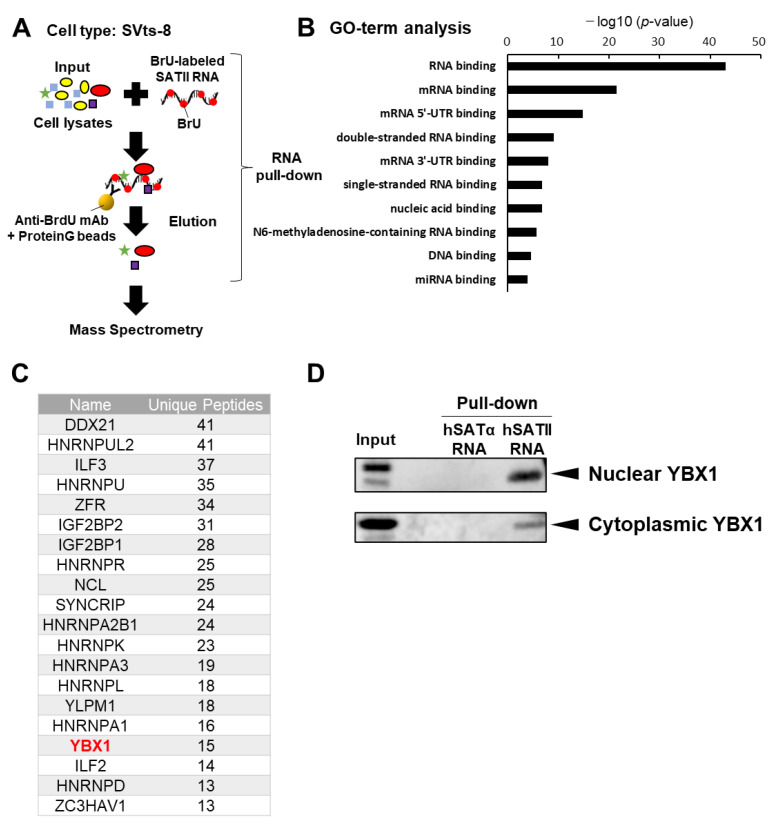
SATII RNA selectively binds to YBX1. (**A**) Overview of sample preparation. RNA pull-down was performed using the whole cell lysates of SVts-8 cells and biosynthesized SATII RNA. The SATII RNA-binding proteins were eluted and analyzed via mass spectrometry. (**B**) Gene ontology (GO) term analysis of SATII RNA binding proteins. (**C**) List of the top 20 RNA-binding proteins identified using the GO term analysis of SATII RNA-binding proteins. (**D**) SVts-8 cells were separated into nuclear and cytoplasmic fractions. Western blot analysis was performed after RNA pull-down by SATα RNA or SATII RNA using each fraction.

**Figure 2 ijms-24-16399-f002:**
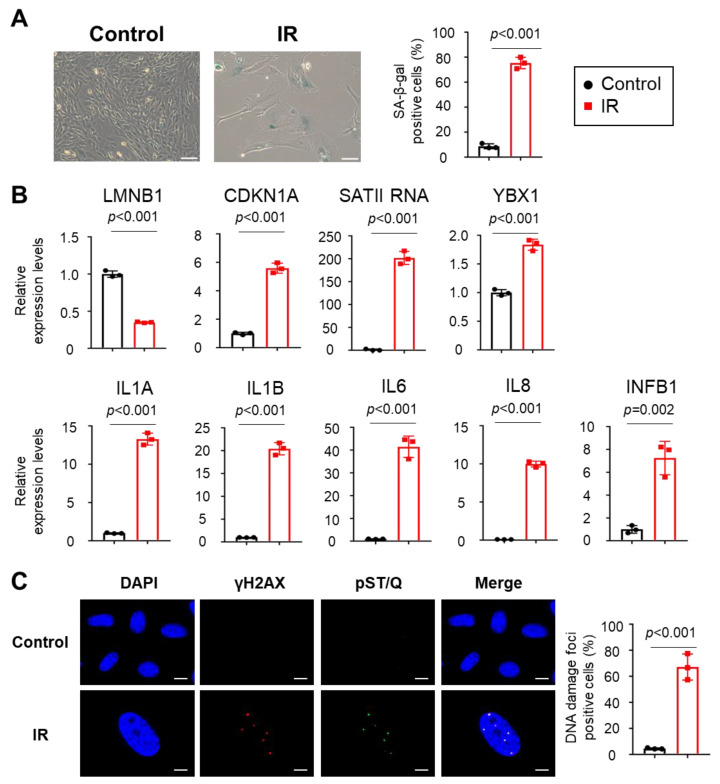
Cellular senescence was induced via X-ray irradiation in RPE-1 cells. (**A**) RPE-1 cells were irradiated at 40 Gy X-ray and incubated for 10 days to induce cellular senescence. Senescence-associated β-galactosidase (SA-β-gal) staining of control and X-ray–induced senescent cells (IR). The bar graphs indicate the percentage of SA-β-gal–positive cells. Results represent the mean ± standard deviation (SD). Scale bar = 100 µm. (**B**) Relative RNA levels of LMNB1, CDKN1A, IL1A, IL1B, IL6, IL8, INFB1, and SATII RNA in control and IR cells. Relative quantitation data represent the mean ± SD normalized to actin β. (**C**) Immunofluorescence staining of DNA damage response markers γH2AX (red), pST/Q (green), and 4′,6-diamidino-2-phenylindole (blue) in control and IR cells. The bar graphs indicate the percentage of nuclei containing more than two positive foci for both γH2AX and pST/Q staining from at least 100 cells per condition for three independent experiments. Scale bar = 10 µm. Results represent the mean ± SD. *p*-Values were calculated via unpaired two-tailed Student’s *t*-test in all panels.

**Figure 3 ijms-24-16399-f003:**
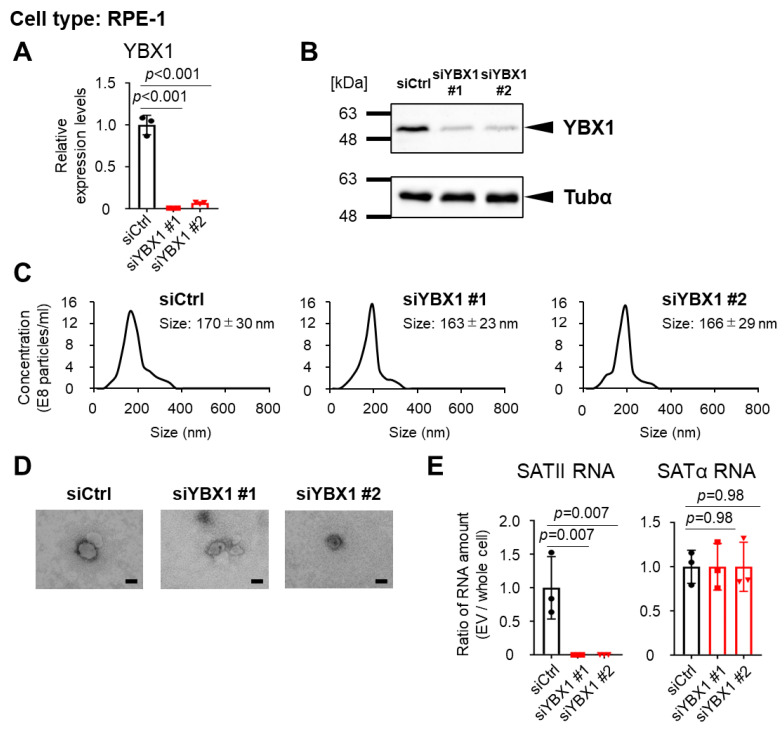
YBX1 knockdown reduced loads of SATII RNA in small extracellular vesicles (sEVs). (**A**) Relative mRNA levels of YBX1 in siRNA control (siCtrl) and YBX1 knockdown RPE-1 cells were measured by reverse transcription-quantitative polymerase chain reaction. Relative quantitation data represent the mean ± standard deviation (SD) normalized to actin β. (**B**) Western blot analysis of whole cell lysate of siCtrl and YBX1 knockdown RPE-1 cells. Tubulinα (Tubα) was used as the loading control. (**C**) Nanoparticle tracking analysis for quantitative measurement of sEVs collected from siCtrl or YBX1 knockdown cells. (**D**) Immunoelectron microscopy with anti-CD63 antibody for measuring the particle size and shape of sEVs collected from YBX1 knockdown cells. Scale bar = 100 nm. (**E**) Ratio of the amount of SATII RNA and SATα RNA contained in sEVs divided by the amount contained in whole cells under treatment with YBX1 or control siRNA. Relative quantitation data represent the mean ± SD. *p*-Values were calculated via one-way analysis of variance with Dunnett’s multiple comparisons test in panels (**A**,**E**).

**Figure 4 ijms-24-16399-f004:**
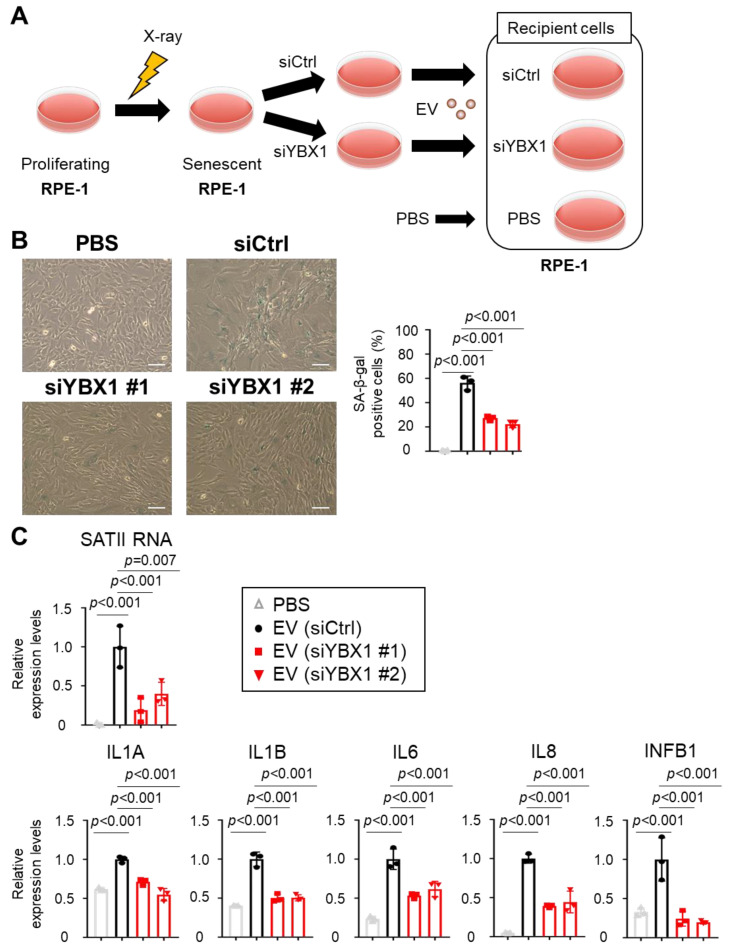
Small extracellular vesicles (sEVs) derived from senescent cells induced cellular senescence to proliferating cells via YBX1. (**A**) Experimental scheme of this analysis. sEVs collected from siRNA control (siCtrl) or YBX1 knockdown conditions were added to proliferating RPE-1 cells. (**B**) Senescence-associated β-galactosidase (SA-β-gal) staining of the cells upon the addition of sEVs collected from siCtrl or YBX1 knockdown cells to proliferating RPE-1 cells. The bar graphs indicate the percentage of SA-β-gal–positive cells. Results represent the mean ± standard deviation (SD). Scale bars = 100 µm. (**C**) Relative RNA levels of SATII RNA, IL1A, IL1B, IL6, IL8, and INFB1 in recipient cells. Relative quantitation data are expressed as mean ± SD normalized to actin β. *p*-Values were calculated via one-way analysis of variance with Dunnett’s multiple comparisons test in panels (**B**,**C**).

**Figure 5 ijms-24-16399-f005:**
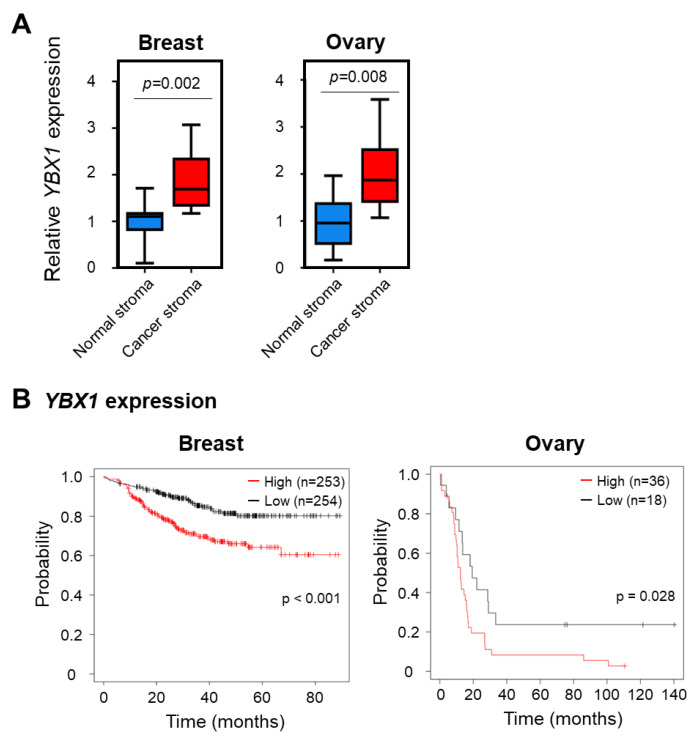
Expression levels of *YBX1* in breast and ovarian cancer tissues. (**A**) *YBX1* mRNA levels in normal and tumor stromal tissues in breast and ovarian cancer specimens. The left boxplot shows *YBX1* mRNA expression in normal breast stroma (*n* = 13) and breast cancer stroma (*n* = 13). The right boxplot shows *YBX1* mRNA expression in normal ovarian stroma (*n* = 8) and ovarian cancer stroma (*n* = 13). *p*-Values were calculated using Wilcoxon test. (**B**) Evaluation of differences in recurrence-free survival associated with different *YBX1* expression levels in breast and ovarian cancer specimens. The clinical data GSE25066 (breast cancer samples) and GSE30161 (ovarian cancer samples) were analyzed using Kaplan–Meier plotter (https://kmplot.com/analysis/, accessed on 5 September 2023).

**Figure 6 ijms-24-16399-f006:**
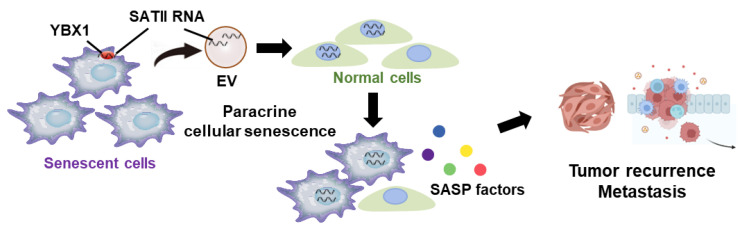
Graphic abstract for this study. YBX1 incorporates SATII RNA into sEVs in senescent cells. These sEVs promote cellular senescence of surrounding recipient cells in the cancer microenvironment. This phenotype may lead to cancer progression such as tumor recurrence and metastasis. This figure was created with BioRender.com.

## Data Availability

The data that support the findings of this study are available in the Appendix A of this article. The Recurrence-free survival prognoses raw files can be accessed via NCBI GEO under record numbers GSE4823 and GSE40595. The raw data utilized in this study will be made publicly available upon publication via the Japan Proteome Standard Repository/Database (jPOST), ID: JPST002366.

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
