# Peer review of "YBX1 Regulates Satellite II RNA Loading into Small Extracellular Vesicles and Promotes the Senescent Phenotype"

_ijms, 2023, doi:10.3390/ijms242216399_

Round 1

Reviewer 1 Report

Comments and Suggestions for Authors

The work by Chiba et al brings out the role of an RNA binding protein, YBX1, in the incorporation of SATII RNA into sEVs and how it influences the senescence phenotype. The concept is interesting and the observations made are quite straight forward. 

The manuscript needs further work before it can be accepted for publication.

The comments are:

1. Introduction: An introduction pertaining to YBX1, its normal physiological functions etc needs to be included to understand why this particular molecule was selected for the study. Also sEVs and its role in senescence can be introduced in a separate paragraph. This would help the reader understand the relevance and importance of the work.

2. What is the rationale for selecting YBX1 while Fig 1C shows hits that are more relevant than YBX1? 

3. What seems confusing is why the initial pull down experiments were done in SVts 8 cells and the subsequent experiments in RPE1 and IMR90?

4. The radiation dose used 40Gy seems to be on the higher side - please provide appropriate references for selecting this dose.

5. Other than the inflammatory markers checked, it would be nice to see the effect on anti inflammatory molecules - probably IL10?

6. What is the n=3 shown in all the figures? Is it the technical replicates or biological replicates? n=3 seems a low number especially for in vitro experiments. 

7. The authors need to show whether inducing senescence increased YBX1 levels. 

8. Figure 4 -the SA-b-gal staining is very faint. Please provide better images. 

Also correct the figure legend from histogram to bar graph/column graphs.

9. Lines 154-156 say: "we found that the number of SA-β-gal–positive cells increased in the cells treated with sEVs collected from X-ray–induced senescent RPE-1 cells compared with the cells treated with phosphate-buffered solution (PBS) (Fig. 4B)" -  please provide the data. As Fig 4B - compares PBS vs siC vs siYBX1, and the above said data is not included. 

10. In the discussion - Line 209 - it is unclear why IL-6 is discussed - and that too in a single line without any follow up. 

11. ImmunoEM methodology missing in Section 4. 

12. Minor typographical errors- like non-cording, NF@b etc. Please correct. 

Comments on the Quality of English Language

Minor typographical errors. 

Reviewer 2 Report

Comments and Suggestions for Authors

1) In the Introduction section,  provide a brief description of YBX1 and it importance to study. 

2) Acknowledge previous study PMID: 29073095 discussing about YBX1 and its role in EVs secretion. It will be introduce how your study is different and further advancing YBX1 role. 

3) Spell check required.

4) Figure 2 A, very less cells in IR group? Do you generally observe this high cell death. 

5) Figure 2C, please provide a scale bar. Magnification looks different between control and IR group. 

6) What happens to the expression of YBX1 after senescence. 

7) Please provide quantification for Figure 3D. 

8) Why was PBS used as control instead of isolating sEVs from untreated cells. 

9) Is there any difference between siYBX1 and siYBX2? 

Comments on the Quality of English Language

Overall ok but spell check required. 

Round 2

Reviewer 1 Report

Comments and Suggestions for Authors

Thanks for addressing the issues.

I recommend the manuscript to be accepted in the present form.